# Strategies to improve sexual health in women with urinary incontinence: A scoping review

Zoleikha Askarloo[1], Shadab Shahali[1]*, Fazlollah Ahmadi[2], Ali Montazeri[3,4]

1 Department of Reproductive Health and Midwifery, Faculty of Medical Sciences, Tarbiat Modares Univesity, Tehran, Iran, 2 Department of Nursing, Faculty of Medical Sciences, Tarbiat Modares University, Tehran, Iran, 3 Health Metrics Research Center, Institute for Health Sciences Research, ACECR, Tehran, Iran, 4 Faculty of Humanity Sciences, University of Science and Culture, Tehran, Iran

* shadab.shahali@modares.ac.ir

## Abstract

### Background

Urinary incontinence (UI), affecting 3%–55% of women worldwide, significantly impairs sexual health and quality of life. Despite its prevalence, interventions targeting sexual dysfunction in this population are understudied. This scoping review synthesizes evidence on strategies to improve sexual health in women with UI, aiming to guide clinical practice and future research.

### Methods

Adhering to Arksey and O'Malley's scoping review framework, we systematically searched PubMed, Scopus, Web of Science, and gray literature (2015–2025) for studies on interventions addressing sexual health in women with UI. Two reviewers independently screened and extracted data, synthesizing findings narratively into four categories: non-surgical interventions, surgical interventions, comparative effectiveness, and partner sexual function.

### Results

The included 31 studies evaluated a broad range of interventions, including pelvic floor muscle training, behavioral therapies, pharmacological treatments, and surgical procedures. Reported effects on sexual function were inconsistent and varied across studies, depending on intervention type, study design, baseline sexual function status, and measurement tools. While some studies reported improvements in specific domains of sexual function following certain interventions, others reported no significant change or mixed outcomes. Evidence regarding partner-related sexual outcomes was limited and based on a small number of studies with modest sample sizes.

**Data availability statement:** All relevant data are derived from previously published studies included in this scoping review and are fully available within the cited articles listed in the reference section. The dataset consists of extracted and charted data from these sources, along with the search strategies described in the Methods section. No primary data were collected from human participants, and no separate dataset is hosted in an external repository.

**Funding:** The author(s) received no specific funding for this work.

**Competing interests:** NO authors have competing interests.

**Abbreviations:** AHRQ, agency for healthcare research and quality; CBT, cognitive-behavioral therapy; ES, electrical stimulation; FSD, female sexual dysfunction; FSFI, female sexual function index; GRISS, golombok rust inventory of sexual satisfaction; ICIQ-SF, international consultation on incontinence questionnaire-short form; ICIQ-UI, international consultation on incontinence questionnaire-urinary incontinence; IIEF-5, international index of erectile function-5; IIQ-7, incontinence impact questionnaire-7; JBI, joanna briggs institute; MSHQ, male sexual health questionnaire; MUI, mixed urinary incontinence; MUT, midurethral sling; NICE, national institute for health and care excellence; NICHSR, national information center on health services research and health care technology; OAB, overactive bladder; PCC, population-concept-context; PFBQ, pelvic floor bother questionnaire; PFDI, pelvic floor distress inventory; PFDI-20, pelvic floor distress inventory-20; PFIQ, pelvic floor impact questionnaire; PFMT, pelvic floor muscle training; PGI-I, Patient Global Impression of improvement; PISQ-12, pelvic organ prolapse/urinary incontinence sexual function questionnaire-12; PISQ-IR, pelvic organ prolapse/urinary incontinence sexual function questionnaire-incontinence and/or prolapse; PMS, pulsed magnetic stimulation; POP, pelvic organ prolapse; POP-Q, pelvic organ prolapse quantification; P-QoL, prolapse quality of life; PRISMA, preferred reporting items for systematic reviews and meta-analyses; RT, retropubic tape; SF-36, short form-36 health survey; SIS, single incision sling; SSEL-W-SF, sexual self-esteem inventory for women-short form; SUI, stress urinary incontinence; TAH,

## Conclusion

This scoping review demonstrates that multiple interventions have been explored for their potential impact on sexual function in women with urinary incontinence; however, the available evidence is heterogeneous and largely descriptive. Due to variability in populations, interventions, and outcome assessment, no definitive conclusions regarding effectiveness or comparative benefit can be drawn. Future research using standardized sexual function measures and clearly defined populations is needed to better inform clinical practice and guideline development.

## Introduction

Urinary incontinence (UI) is a common and distressing condition defined by the International Urogynecological Association as any involuntary leakage of urine resulting in social or health-related discomfort [1]. Although it affects both sexes, UI is more prevalent among women due to anatomical, hormonal, and obstetric factors. Global prevalence estimates range from 3% to 55% and increase substantially with age [2].

UI is classified into stress urinary incontinence (SUI), urgency urinary incontinence (UUI), mixed urinary incontinence (MUI), and overflow incontinence. SUI involves leakage during increased intra-abdominal pressure and is commonly associated with urethral hypermobility or intrinsic sphincter deficiency, whereas UUI is characterized by involuntary leakage accompanied by urgency and is often linked to detrusor overactivity [3,4]. MUI presents features of both, while overflow incontinence, the least common subtype in women, results from bladder overdistension due to obstruction or neurological impairment [5].

UI represents a significant public health concern because of its high prevalence, economic burden, and negative impact on physical and psychological well-being [6]. Increasing evidence suggests that UI is associated with female sexual dysfunction (FSD), likely due to the close anatomical and functional relationship between the urinary and reproductive systems. However, research examining this association remains limited and methodologically heterogeneous, resulting in inconsistent findings [7]. Coital incontinence, in particular, is a highly distressing symptom that substantially impairs sexual satisfaction and intimacy [8].

Beyond physical symptoms, UI is associated with adverse mental health outcomes, including depressive symptoms, reduced self-esteem, and social withdrawal [9,10]. These multidimensional consequences underscore the importance of incorporating sexual health considerations into patient-centered UI care [11]. Current management strategies include conservative, pharmacological, and surgical approaches [12]; however, sexual health outcomes are not consistently evaluated when assessing treatment effectiveness [13].

Despite growing recognition of the relationship between UI and sexual health, substantial gaps persist in sexual health service provision for affected women. Healthcare providers frequently report limited knowledge or discomfort in addressing sexual concerns related to UI, contributing to stigma and unmet needs [14]. These

total abdominal hysterectomy; TBSF, transvaginal bilateral sacrospinous fixation; TLH, total laparoscopic hysterectomy; TMUS, transobturator midurethral sling; TOT, transobturator tape; TOA/TVA, tension-free obturator/transvaginal adjustable; TVT, tension-free vaginal tape; UDI-6, urogenital distress inventory-6; UI, urinary incontinence; UUI, urgency urinary incontinence; VAS, visual analogue scale; VH, vaginal hysterectomy; VNTR, vaginal native tissue repair; VVP, vaginal vault prolapse.

challenges are amplified in conservative cultural settings where discussions of sexual health are particularly sensitive [15,16]. Furthermore, there is a lack of structured, evidence-based strategies specifically aimed at improving sexual health outcomes in women with UI [15].

Existing literature has largely focused on UI symptom management, with comparatively little attention given to interventions targeting sexual well-being [17]. Previous systematic reviews, including that by Pinheiro Sobreira Bezerra et al. (2020), have primarily examined prevalence and associations between UI and sexual function rather than systematically mapping intervention strategies [7].

The aim of this scoping review was to systematically map and describe the available evidence on surgical and non-surgical interventions evaluated for improving sexual health–related outcomes, primarily sexual function, in women with urinary incontinence, and to identify key knowledge gaps, including the limited evidence on partner-related sexual outcomes.

## Methods

This scoping review was conducted following the methodological framework proposed by Arksey and O'Malley [18], with enhancements based on recommendations from the Joanna Briggs Institute (JBI) for scoping reviews [19]. The completed PRISMA-ScR checklist is provided as Supplementary file 2.

### Formulating the research question

To ensure a comprehensive exploration of the topic, the research question was developed using the Population-Concept-Context (PCC) framework: *What is the impact of interventions for urinary incontinence on sexual health outcomes, and how do interventions focused on sexual function compare to those primarily targeting urinary incontinence?"*

- **Population (P):** Women with urinary incontinence

- **Concept (C):** Strategies or interventions aimed at improving sexual health (Although sexual health is a multidimensional construct encompassing physical, psychological, and relational aspects of sexuality, the outcomes synthesized in this review primarily reflect sexual function as measured by validated questionnaires, which are commonly used proxies for sexual health–related outcomes in clinical research.)

- **Context (C):** Healthcare settings, including clinics, hospitals, and specialized health centers

### Data sources and search strategy

A comprehensive literature search was conducted in scientific databases such as PubMed, Scopus, Web of Science, ScienceDirect, the World Health Organization (WHO) repository, the National Institute for Health and Care Excellence (NICE), and the Agency for Healthcare Research and Quality (AHRQ) to identify relevant studies.

The search strategy incorporated controlled vocabulary (e.g., MeSH terms) and relevant keywords were: Sexual Health, Sexual Health Improvement, Reproductive Health, Sexual Health Provider, Urinary Incontinence, Stress Urinary Incontinence, Urge Urinary Incontinence, Overflow Incontinence, Mixed Urinary Incontinence, Sexual Health Service. Sexual health–related terms were intentionally broadened to include dyspareunia, libido, sexual desire, arousal, and vaginal pain to minimize the risk of missing relevant studies.

The search was restricted to peer-reviewed English articles (quantitative, qualitative, mixed-methods studies, and review articles) published between 2015 and 2025. (The search strategy is detailed in Table 1.) The literature search was limited to studies published between January 2015 and December 2025. This timeframe was selected to capture evidence reflecting contemporary clinical practice in the management of urinary incontinence, including modern surgical techniques (e.g., current mid-urethral sling approaches), updated pelvic floor muscle training (PFMT) protocols, and the widespread use of validated sexual function instruments such as the FSFI and PISQ-12 with standardized scoring and interpretation. Given substantial changes in intervention techniques and outcome measurement over the past decade, restricting the timeframe was intended to enhance the clinical relevance and interpretability of the findings.

Additionally, gray literature was considered, including books, book chapters, government reports, clinical guidelines, and policy documents from recognized institutions such as the National Information Center on Health Services Research and Health Care Technology (NICHSR).

### Eligibility criteria

Studies were included if they involved adult women (≥18 years) diagnosed with any type of urinary incontinence and assessed sexual health outcomes (e.g., sexual function, satisfaction, or partner-related factors). Studies focusing on neurological populations or men were excluded. Partner-related outcomes were considered only when reported alongside female participants.

### Study selection

Two independent reviewers screened the titles and abstracts using a standardized selection tool. Studies meeting inclusion criteria underwent a full-text review, followed by independent double screening. A third reviewer resolved discrepancies. Studies were assessed for quality using the Mixed Methods Appraisal Tool (version 2018). Although quality appraisal is not routinely used to exclude studies in scoping reviews, the MMAT was applied in this review to support its role as a preparatory phase for future guideline development in sexual health among women with urinary incontinence. To ensure that mapped intervention strategies were derived from methodologically sound evidence, studies scoring below 3/5 were excluded due to substantial limitations in study design, outcome measurement, or reporting quality. Fig 1 presents the PRISMA flow diagram of the selection process [20].

### Data extraction and synthesis

A standardized data extraction form was used to systematically collect study characteristics, including study design, population, type of UI, interventions, and key findings. Two researchers independently extracted data, and inconsistencies were resolved through discussion.

Sexual function outcomes were extracted as reported in the primary studies using validated questionnaires such as the Female Sexual Function Index (FSFI) and the Pelvic Organ Prolapse/Urinary Incontinence Sexual Questionnaire (PISQ). Baseline sexual dysfunction status was not consistently reported across studies; therefore, changes in sexual function scores were descriptively synthesized without assuming the presence of clinically diagnosed sexual dysfunction at study entry. No uniform thresholds for clinically significant change were applied, as these were rarely defined in the original studies.

**Table 1. Search Strategy.**

| PubMed | ("Urinary Incontinence"[Mesh] OR |
|---|---|
| | "Urinary Incontinence, Stress"[Mesh] OR |
| | "Urinary Incontinence, Urge"[Mesh] OR |
| | "Mixed Urinary Incontinence"[All Fields] OR |
| | "Overactive Bladder"[Mesh] OR |
| | "urinary incontinence"[tiab] OR |
| | "stress urinary incontinence"[tiab] OR |
| | "urge urinary incontinence"[tiab] OR |
| | "mixed urinary incontinence"[tiab] OR |
| | "coital incontinence"[tiab]) |
| | AND |
| | ("Sexual Health"[Mesh] OR |
| | "Sexual Function"[Mesh] OR |
| | "Sexual Dysfunction, Physiological"[Mesh] OR |
| | "Sexual Dysfunction, Psychological"[Mesh] OR |
| | "Dyspareunia"[Mesh] OR |
| | "Libido"[Mesh] OR |
| | "sexual function"[tiab] OR |
| | "sexual dysfunction"[tiab] OR |
| | "sexual health"[tiab] OR |
| | "sexual satisfaction"[tiab] OR |
| | "sexual desire"[tiab] OR |
| | "sexual arousal"[tiab] OR |
| | "libido"[tiab] OR |
| | "dyspareunia"[tiab] OR |
| | "vaginal pain"[tiab]) |
| | AND |
| | ("Therapeutics"[Mesh] OR |
| | "Treatment Outcome"[Mesh] OR |
| | "Pelvic Floor Muscle Training"[All Fields] OR |
| | "Physical Therapy Modalities"[Mesh] OR |
| | "Behavior Therapy"[Mesh] OR |
| | "Cognitive Behavioral Therapy"[Mesh] OR |
| | "Surgical Procedures, Operative"[Mesh] OR |
| | "Midurethral Sling"[All Fields] OR |
| | "Anticholinergic Agents"[Mesh] OR |
| | "Magnetic Stimulation Therapy"[Mesh] OR |
| | intervention*[tiab] OR |
| | treatment*[tiab] OR |
| | therap*[tiab] OR |
| | surgery[tiab]) |
| | AND |
| | (    "2015/01/01"[Date – Publication]: "2025/12/31"[Date – Publication]) |
| | AND |
| | (English[lang]) |

*(Continued)*

**Table 1.** (Continued)

| | |
|---|---|
| Scopus | TITLE-ABS-KEY |
| | ("urinary incontinence" OR |
| | "stress urinary incontinence" OR |
| | "urge urinary incontinence" OR |
| | "mixed urinary incontinence" OR |
| | "overactive bladder" OR |
| | "coital incontinence") |
| | AND |
| | TITLE-ABS-KEY |
| | (  "sexual health" OR |
| | "sexual function" OR |
| | "sexual dysfunction" OR |
| | "sexual satisfaction" OR |
| | "sexual desire" OR |
| | "sexual arousal" OR |
| | libido OR |
| | dyspareunia OR |
| | "vaginal pain") |
| | AND |
| | TITLE-ABS-KEY |
| | (   intervention* OR |
| | treatment* OR |
| | therap* OR |
| | surgery OR |
| | "pelvic floor muscle training" OR |
| | PFMT OR |
| | "pulsed magnetic stimulation" OR |
| | pessary OR |
| | "midurethral sling" OR |
| | CBT OR |
| | anticholinergic*) |
| | AND |
| | (PUBYEAR > 2014) |
| | AND |
| | (LIMIT-TO (LANGUAGE, "English")) |
| Web of Science | TS= |
| | ("urinary incontinence" OR |
| | "stress urinary incontinence" OR |
| | "urge urinary incontinence" OR |
| | "mixed urinary incontinence" OR |
| | "overactive bladder" OR |
| | "coital incontinence") |
| | AND |
| | TS= |
| | ("sexual health" OR |
| | "sexual function" OR |
| | "sexual dysfunction" OR |

*(Continued)*

**Table 1.** (Continued)

| | |
|---|---|
| | "sexual satisfaction" OR |
| | "sexual desire" OR |
| | "sexual arousal" OR |
| | libido OR |
| | dyspareunia OR |
| | "vaginal pain") |
| AND | |
| TS= | |
| (intervention* OR | |
| | treatment* OR |
| | therap* OR |
| | surgery OR |
| | "pelvic floor muscle training" OR |
| | PFMT OR |
| | "pulsed magnetic stimulation" OR |
| | pessary OR |
| | "midurethral sling" OR |
| | CBT OR |
| | anticholinergic*) |
| Refined by: | |
| - Publication Years:   2015–2025 | |
| - Languages: English | |

Given the heterogeneity of study designs, interventions, and outcome measures, the researchers employed a narrative data Synthesis approach as described by Popay et al. [21].

The extracted data were compiled into summary tables, detailing study characteristics, target populations, UI subtypes, intervention strategies, and reported outcomes. Common themes were identified and compared across studies to generate a comprehensive understanding of the strategies used to improve sexual health in women with UI.

**Human Ethics and Consent to Participate declarations:** This study did not involve human participants or identifiable personal data and therefore did not require institutional ethics approval.

## Results

A total of 31 studies were included in this review, examining various interventions aimed at improving sexual health in women with UI. The studies varied in design, population characteristics, and intervention types, including behavioral, surgical, pharmacological, and psychological approaches. Where data permitted, findings were stratified according to underlying condition (stress urinary incontinence, urgency urinary incontinence, mixed urinary incontinence, or pelvic organ prolapse).

### Study characteristics

The studies cover a wide range of interventions, including surgical procedures such as transobturator tape (TOT), midurethral sling (MUT), tension-free vaginal tape (TVT), native tissue repair, vaginal hysterectomy, sacrospinous hysteropexy, and mesh-based repairs, as well as non-surgical approaches like pulsed magnetic stimulation (PMS), pessary use, and cognitive-behavioral therapy (CBT). The populations predominantly consist of sexually active women, though some studies include broader groups with pelvic floor disorders, including those with POP stages II–IV, uterine prolapse, or

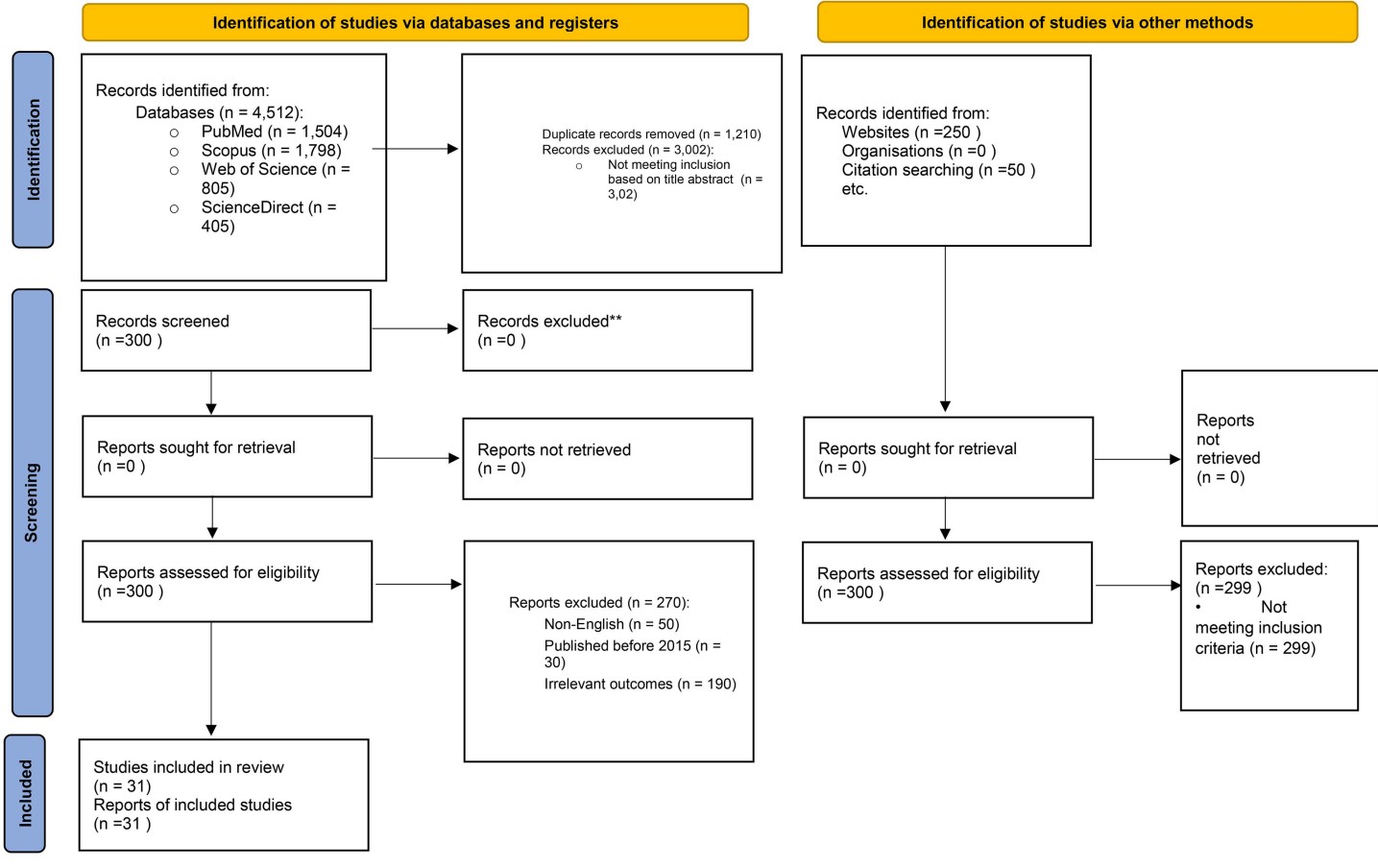

**Fig 1. PRISMA 2020 flow diagram.**

coital incontinence. Sample sizes vary widely, from small cohorts (e.g., 20 women in Vitale et al., 2018) to large registries (e.g., 1210 women in Solhaug et al., 2024) [22], reflecting diverse methodological approaches. The majority of the studies utilized validated questionnaires, such as the Pelvic Organ Prolapse/Urinary Incontinence Sexual Function Questionnaire and the Female Sexual Function Index, to assess changes in sexual health outcomes. The follow-up periods varied from a few weeks to over 10 years, providing insights into both short-term and long-term effects of different interventions. Supplementary file1summarizes the key findings of the included studies

## Analysis of the studies

Results were categorized into four main groups: Impact of non-surgical Interventions on sexual function, Effectiveness of Surgical Interventions on sexual function, Comparative Outcomes of Surgical vs. Non-Surgical Approaches, and Partner Sexual Function (Reported changes in sexual function scores reflect study-specific findings and should not be interpreted as treatment effects for established sexual dysfunction).

## Impact of non-surgical Interventions on sexual function

Non-surgical strategies, including pelvic floor muscle training (PFMT), electrical stimulation (ES), pulsed magnetic stimulation (PMS), cognitive-behavioral therapy (CBT), pessary use, and anticholinergics, demonstrate a significant potential to

enhance sexual function in women with UI. PFMT, as a cornerstone of conservative management, consistently improves sexual function by enhancing pelvic muscle tone and sensation, as evidenced by Jha et al. [23] in their RCT. The lack of difference between standard PFMT and PFMT with ES suggests that additional technologies may not always yield superior outcomes, though ES's cost-effectiveness [23,24] highlights its practical utility in resource-limited settings. Similarly, PMS emerges as a novel and effective option, with Studies Lim et al. and González-Isaza et al. [25,26] reporting significant improvements in sexual function and UI symptoms over short follow-up periods, underpinned by high patient satisfaction. CBT [27] uniquely addresses psychological dimensions, enhancing sexual self-esteem alongside function, suggesting a holistic approach for women with UI-related distress. Pessary use [28] maintains sexual function while improving UI symptoms, offering a non-invasive alternative for those avoiding surgery. While anticholinergics can reduce OAB urgency, they don't improve sexual function, potentially due to psychosomatic factors or side effects like dryness [23]. Evidence for anticholinergic efficacy remains inconclusive, primarily stemming from a small study without urodynamic confirmation of detrusor overactivity. Collectively, these findings underscore the importance of non-surgical strategies as first-line management for women with UI, particularly those seeking minimally invasive options. However, the variability in treatment efficacy emphasizes the necessity for personalized therapeutic approaches tailored to individual patient needs and clinical profiles.

## Effectiveness of surgical Interventions

Surgical interventions for UI, including midurethral slings (MUTs), colposuspension, fascial slings, and prolapse repairs, demonstrate significant efficacy in improving sexual function—primarily by mitigating coital incontinence, a major impediment to sexual health in affected women. A study [23] revealed that approximately one-third of women undergoing continence surgery report improved sexual function, while 55.5% experience no change and 13.1% note deterioration, highlighting the heterogeneous outcomes of these interventions.

Among surgical options, MUTs—such as tension-free vaginal tape (TVT) and transobturator tape (TOT)—are particularly effective, with studies [22,29–34] documenting sustained benefits over follow-up periods ranging from 12 months to 20 years. These procedures not only restore continence but also enhance sexual satisfaction, likely attributable to regained confidence and reduced leakage during intercourse. Modified colposuspension [35] and native tissue repairs [36,37] similarly yield high patient satisfaction and durable symptom relief, as evidenced by long-term follow-up data. However, outcomes vary across techniques. Fascial slings are associated with higher rates of dyspareunia [38], while some MUT recipients report worsened sexual function due to postoperative pain or altered sensation. Advanced surgical approaches, such as adjustable mesh systems [39] and laparoscopic sacrocolpopexy [40], show promise in further optimizing outcomes with low complication rates, suggesting that technological refinements contribute to improved efficacy. Some studies comparing different hysterectomy types found that hysterectomy, regardless of the technique (vaginal, abdominal, laparoscopic, or with sacrospinous hysteropexy), generally does not negatively impact overall sexual function, and can even improve it by resolving prolapse-related issues.Radical hysterectomy may have a more detrimental effect on sexual function and pelvic floor issues compared to simple hysterectomy. While vaginal length may be affected by the surgical approach, it doesn't seem to have a significant impact on sexual function [41–44].

Weight loss surgery also demonstrates a positive impact on both pelvic floor disorders and sexual performance in a single study [45].

Collectively, surgical interventions remain a cornerstone for severe UI, offering robust anatomical correction and symptom resolution. Nevertheless, the potential for adverse sexual outcomes underscores the importance of meticulous patient selection, individualized counseling, and consideration of less invasive alternatives where appropriate.

## Comparative outcomes of surgical vs. non-surgical approaches

Comparative analyses demonstrate a clear dichotomy between surgical and non-surgical interventions for urinary incontinence (UI), with each approach offering distinct advantages and limitations. Surgical interventions [23,32,46,47] consistently

demonstrate superior efficacy in anatomical restoration and reduction of coital incontinence, with emerging techniques such as mini-slings and autologous transobturator tape (TOT) showing improved safety profiles compared to traditional midurethral slings (MUTs). These procedures often provide durable, long-term benefits, making them particularly suitable for patients with severe symptoms or those who have failed conservative management. In contrast, non-surgical modalities—including pelvic floor muscle training (PFMT) [23], pulsed magnetic stimulation (PMS) [26], and cognitive-behavioral therapy (CBT) [27]—achieve comparable improvements in sexual function while avoiding surgical risks. These approaches are particularly advantageous for patients with milder symptoms or those prioritizing minimally invasive treatment. Pessary use [28] demonstrates efficacy similar to minor surgical interventions in preserving sexual function, whereas anticholinergic therapy [23] shows limited benefits, underscoring its condition-specific limitations. Further examination of surgical techniques [48] suggests that native tissue repairs may offer an optimal balance between efficacy and safety compared to mesh-based alternatives. These findings support a stepped-care approach, with non-surgical interventions serving as first-line therapy, followed by surgical options for non-responders or severe cases. However, the current evidence base is constrained by a paucity of direct head-to-head comparative trials, necessitating individualized treatment decisions based on symptom severity, patient comorbidities, and personal preferences. (Table 2 shows the pros and cons of interventions for improving the sexual health of women with UI.)

**Partner sexual function**

Only three studies assessed partner-related sexual outcomes following interventions for urinary incontinence. Across these studies, with partner sample sizes that were either small and evaluated using heterogeneous outcome measures. Due to the limited number of studies, small sample sizes, and variability in outcome assessment, findings related to partner sexual outcomes should be interpreted with caution.

The impact of UI interventions on partners' sexual function remains inadequately investigated in the current literature. Available evidence suggests that while treatment yields significant improvements in female sexual function [29,49–52], these benefits appear largely patient-specific, with no corresponding enhancement in male partners' sexual satisfaction or function as measured by standardized instruments (MSHQ, IIEF-5).

Notably, Jha et al. [23] observed higher rates of partner avoidance in cases of pelvic organ prolapse (POP) compared to SUI, suggesting condition-specific variations in relational dynamics. However, detailed assessments of partner outcomes were not reported. Similarly, while one study [25] implied improved couple-level satisfaction following pulsed magnetic stimulation (PMS), it lacked specific metrics evaluating partners' sexual function.

The limited focus on partner outcomes reflects a broader gap in understanding sexual mutuality, where both partners' experiences shape relational satisfaction. Jha et al. [23] noted higher partner avoidance in POP compared to SUI, suggesting that condition-specific symptoms influence interpersonal dynamics differently. The absence of validated partner-focused metrics, such as the International Index of Erectile Function (IIEF-5), in most studies limits conclusions about mutual sexual satisfaction. Qualitative insights, though scarce, suggest that shame and avoidance behaviors in women with UI may indirectly affect partner intimacy, warranting further exploration.

**Table 2. Pros and Cons of Interventions for Improving Sexual Health in Women with UI.**

| Intervention Category | Pros | Cons |
| --- | --- | --- |
| Non-Surgical (PFMT, PMS, CBT, Pessary) | Minimally invasive, cost-effective, improves sexual self-esteem (CBT), high patient acceptability | Variable efficacy, requires patient compliance, limited impact on severe UI |
| Surgical (MUT, Native Tissue Repair) | Durable symptom relief, reduces coital incontinence, effective for severe UI | Risk of complications (e.g., dyspareunia), 13% deterioration in sexual function, higher cost |

These findings collectively indicate that UI interventions primarily address female sexual dysfunction, with any partner benefits likely being indirect (e.g., through enhanced intimacy) rather than reflecting measurable improvements in sexual function. The current research landscape reveals a significant gap in understanding the interpersonal consequences of UI—including shame, avoidance behaviors, and altered relationship dynamics—which may substantially influence overall relationship quality. Future studies should incorporate comprehensive partner assessments to better elucidate these multidimensional effects.

## Discussion

This scoping review synthesizes evidence on strategies to enhance sexual health in women with UI, addressing a critical yet underexplored intersection of urogynaecology and sexual medicine. The findings illuminate a range of effective interventions, both surgical and non-surgical, that significantly improve sexual function, underscoring the importance of integrating sexual health into UI management to optimize patient-centered outcomes.

The review identified 31 studies. Improvements in sexual function outcomes were reported inconsistently across studies and varied by intervention type and outcome measure. The heterogeneity observed across populations, interventions, and outcome measures should not be interpreted as a lack of scientific contribution. Instead, it reflects the current state of sexual health research in the context of urinary incontinence, where sexual function is often treated as a secondary or poorly defined outcome. By systematically documenting this heterogeneity, the present review clarifies why existing evidence fails to support definitive clinical conclusions and identifies specific methodological priorities for future research.

Studies assessing pelvic floor muscle training (PFMT) demonstrated heterogeneous effects on sexual function. While some randomized controlled trials reported improvements in FSFI domains such as desire and satisfaction, other studies observed no statistically significant change following intervention. PFMT and PMS emerged as particularly robust, enhancing pelvic muscle tone and sexual satisfaction, while CBT addressed psychological barriers, improving sexual self-esteem [23,25–28,45]. Some studies about surgical interventions, including midurethral slings (MUT) and native tissue repairs, show enhanced sexual function and reduced coital incontinence, but some studies show sexual function deterioration [22,23,29–31,35–40]. Non-surgical approaches were comparable to surgery for milder UI, offering less invasive options. However, interventions had minimal impact on partner sexual function, highlighting a patient-centric focus. These findings provide a comprehensive evidence base for tailored treatment strategies based on UI severity and patient preferences.

The results align with prior studies that highlight the efficacy of PFMT as a cornerstone of conservative UI management. For instance, Bø (2020) emphasized PFMT's role in strengthening pelvic floor muscles, corroborating this review's findings on its impact on sexual function through enhanced muscle tone and sensation [5]. Another study showed the effects of PFMT on sexual satisfaction [53]. Similarly, the effectiveness of PMS aligns with emerging literature on neuromodulation techniques [26], which suggest non-invasive alternatives to surgery. Despite the established role of anticholinergics in relieving urgency symptoms [3], their limited efficacy for sexual function, possibly due to side effects like vaginal dryness, requires further investigation. Moreover, evidence supporting anticholinergics is inconclusive, primarily stemming from a small study without urodynamic confirmation of detrusor overactivity.

Surgical outcomes, particularly for MUT, are consistent with long-term data from Solhaug et al. (2024) [22], which reported sustained sexual function improvements over 10–20 years. However, the 13% deterioration rate noted in this review echoes concerns raised by Clark et al. (2020) [38] about postoperative complications like dyspareunia, particularly with fascial slings. The comparable efficacy of non-surgical and surgical interventions for milder UI supports a stepped-care model advocated by Sussman et al. (2020) [17], though discrepancies in partner outcomes highlight a gap not adequately addressed in prior reviews [7]. This review's focus on partner sexual function, albeit limited, adds a novel dimension, revealing a lack of direct benefits for partners, which contrasts with assumptions of relational improvements post-treatment.

The minimal impact on partner sexual function underscores a patient-centric bias in UI research, neglecting sexual mutuality—the reciprocal interplay of satisfaction and intimacy in relationships. This gap may stem from cultural stigmas around discussing sexual health, particularly in conservative societies [15], or methodological challenges in capturing partner perspectives. Future studies should integrate frameworks like sexual systems theory to assess how UI interventions influence couple dynamics, potentially improving relational outcomes through enhanced communication and intimacy.

Rather than providing practice-changing guidance, the findings of this scoping review highlight important gaps and inconsistencies that currently limit the translation of sexual health outcomes into routine urinary incontinence care, emphasizing the need to integrate sexual health assessments into routine UI management. Clinicians may use e non-surgical interventions such as PFMT and PMS as first-line treatments for women with mild to moderate UI, given their safety and patient acceptability. For severe cases, surgical options like MUT offer durable benefits but require careful patient selection and preoperative counseling to mitigate risks of sexual dysfunction. The limited impact on partner sexual function suggests that clinicians can explore couple-based interventions to address relational dynamics, potentially incorporating CBT to enhance intimacy and communication.

From a clinical sexual health perspective, the primary implication of these findings is the need for more structured and consistent assessment of sexual health in women with urinary incontinence, rather than immediate modification of treatment algorithms.

From a policy perspective, healthcare systems should invest in training providers to address sexual health confidently, particularly in conservative societies where stigma may deter open discussion [15]. Guidelines, such as those from NICE or AHRQ, could be updated to include sexual health as a core outcome in UI management, ensuring holistic care. For patients, these findings empower informed decision-making, offering a spectrum of interventions tailored to individual needs and preferences, ultimately enhancing quality of life.

## Strengths of the review

The review's robust methodology, grounded in Arksey and O'Malley's framework and enhanced by JBI recommendations, ensures a systematic and transparent approach. The comprehensive search strategy, spanning PubMed, Scopus, Web of Science, and gray literature from 2015 to 2025, captures a broad evidence base, including diverse interventions and populations. The inclusion of validated tools like the Pelvic Organ Prolapse/Urinary Incontinence Sexual Function Questionnaire (PISQ-12) and Female Sexual Function Index (FSFI) strengthens the reliability of sexual health outcomes. The narrative synthesis approach effectively accommodates the heterogeneity of study designs, providing a cohesive overview of a complex field.

## Limitations

Despite its strengths, the review faces limitations that temper its generalizability.

Considerable heterogeneity existed across studies regarding study design, urinary incontinence subtypes, interventions, and outcome measures, which limited direct comparison and precluded conclusions about treatment effectiveness. However, this heterogeneity also reflects fundamental gaps in the current evidence base.

Sexual dysfunction was not consistently assessed or reported at baseline, and many studies included sexually active women without clearly defining the presence or severity of dysfunction. Consequently, changes in sexual function scores may not necessarily indicate improvement from a clinically defined condition.

In addition, the application and interpretation of sexual function questionnaires (e.g., FSFI, PISQ) varied widely, with limited clarity regarding clinically meaningful change. Grouping biologically distinct conditions such as stress urinary incontinence, urgency urinary incontinence, and pelvic organ prolapse further complicates interpretation of sexual health outcomes and limits the validity of comparisons between surgical and non-surgical interventions.

Finally, evidence on partner-related sexual outcomes was scarce and based on small samples, highlighting an important gap in the literature. Despite these limitations, this review clarifies methodological inconsistencies and identifies priorities for future sexual health–focused research in women with urinary incontinence.

### Future research directions

Most interventions included in this review primarily targeted urinary incontinence management rather than sexual well-being directly. However, many of them produced secondary benefits on sexual health. Future interventions should explicitly integrate couple-based or partner-inclusive strategies to address both physiological and relational domains of sexual health.

Future research should prioritize head-to-head comparative trials of surgical versus non-surgical interventions, standardizing outcome measures (e.g., PISQ-12) and extending follow-ups to assess long-term durability. Investigating partner sexual function using validated tools (e.g., IIEF-5) and qualitative methods could elucidate relational dynamics, addressing the current patient-centric bias. Exploring novel modalities, such as laser therapy or nerve-sparing surgical techniques, may further minimize adverse sexual outcomes. Larger, multicenter studies in diverse populations are needed to ensure broadly applicable findings, particularly in underrepresented regions. Integrating psychological interventions like CBT with physiotherapy could enhance holistic care, especially for urgency UI, where pharmacological options underperform.

A stepped-care model, starting with non-surgical interventions (e.g., PFMT, PMS) for mild to moderate UI and escalating to surgical options (e.g., MUT) for severe cases, can optimize sexual health outcomes while minimizing risks.

### Conclusion

This scoping review significantly advances the understanding of strategies to improve sexual health in women with UI, highlighting the efficacy of both surgical and non-surgical interventions and the need for tailored, patient-centered approaches. By addressing critical gaps in partner outcomes and comparative evidence, it lays the foundation for future research to optimize holistic care. Ultimately, integrating sexual health into UI management is not only feasible but essential, promising enhanced quality of life and well-being for women worldwide.

### Supporting information

**S1 File. Data extraction table of included studies.** (Characteristics of included studies, study design, population, intervention type, sexual health outcomes, and key findings).
(DOCX)

**S2 File. PRISMA-ScR checklist.**
(DOCX)

### Acknowledgments

This research is the outcome of a project at the Tarbiat Modares University. We appreciate the staff of Tarbiat Modares University.

### Author contributions

**Conceptualization:** Shadab Shahali, Fazlollah Ahmadi, Ali Montazeri.

**Data curation:** Shadab Shahali, Zoleikha Askarloo, Ali Montazeri.

**Formal analysis:** Zoleikha Askarloo.

**Investigation:** Zoleikha Askarloo.

**Methodology:** Shadab Shahali, Ali Montazeri.

**Project administration:** Fazlollah Ahmadi, Ali Montazeri.

**Supervision:** Shadab Shahali, Fazlollah Ahmadi.

**Validation:** Fazlollah Ahmadi, Ali Montazeri.

**Writing – original draft:** Shadab Shahali, Zoleikha Askarloo.

**Writing – review & editing:** Shadab Shahali, Zoleikha Askarloo, Fazlollah Ahmadi, Ali Montazeri.

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
