## [Decision Letter · Decision Letter 0]

15 Jan 2026

PONE-D-25-66253Strategies to improve sexual health in women with urinary incontinence: A scoping reviewPLOS One

Dear Dr. Shahali,

Thank you for submitting your manuscript to PLOS ONE. After careful consideration, we feel that it has merit but does not fully meet PLOS ONE’s publication criteria as it currently stands. Therefore, we invite you to submit a revised version of the manuscript that addresses the points raised during the review process.

We look forward to receiving your revised manuscript.

Kind regards,

Richard Kao Lee, M.D.

Academic Editor

PLOS One

Journal Requirements:

4. Please include captions for your Supporting Information files at the end of your manuscript, and update any in-text citations to match accordingly. Please see our Supporting Information guidelines for more information: http://journals.plos.org/plosone/s/supporting-information....

Reviewer's Responses to Questions

**Comments to the Author**

1. Is the manuscript technically sound, and do the data support the conclusions?

Reviewer #1: Yes

Reviewer #2: No

2. Has the statistical analysis been performed appropriately and rigorously? 

Reviewer #1: Yes

Reviewer #2: N/A

3. Have the authors made all data underlying the findings in their manuscript fully available?

The PLOS Data policy requires authors to make all data underlying the findings described in their manuscript fully available without restriction, with rare exception (please refer to the Data Availability Statement in the manuscript PDF file). The data should be provided as part of the manuscript or its supporting information, or deposited to a public repository. For example, in addition to summary statistics, the data points behind means, medians and variance measures should be available. If there are restrictions on publicly sharing data—e.g. participant privacy or use of data from a third party—those must be specified.requires authors to make all data underlying the findings described in their manuscript fully available without restriction, with rare exception (please refer to the Data Availability Statement in the manuscript PDF file). The data should be provided as part of the manuscript or its supporting information, or deposited to a public repository. For example, in addition to summary statistics, the data points behind means, medians and variance measures should be available. If there are restrictions on publicly sharing data—e.g. participant privacy or use of data from a third party—those must be specified.requires authors to make all data underlying the findings described in their manuscript fully available without restriction, with rare exception (please refer to the Data Availability Statement in the manuscript PDF file). The data should be provided as part of the manuscript or its supporting information, or deposited to a public repository. For example, in addition to summary statistics, the data points behind means, medians and variance measures should be available. If there are restrictions on publicly sharing data—e.g. participant privacy or use of data from a third party—those must be specified.requires authors to make all data underlying the findings described in their manuscript fully available without restriction, with rare exception (please refer to the Data Availability Statement in the manuscript PDF file). The data should be provided as part of the manuscript or its supporting information, or deposited to a public repository. For example, in addition to summary statistics, the data points behind means, medians and variance measures should be available. If there are restrictions on publicly sharing data—e.g. participant privacy or use of data from a third party—those must be specified.

Reviewer #1: Yes

Reviewer #2: Yes

4. Is the manuscript presented in an intelligible fashion and written in standard English?

Reviewer #1: Yes

Reviewer #2: Yes

5. Review Comments to the Author

Reviewer #1: The topic is important and clinically relevant. Reviewing sexual function in women with urinary incontinence is useful, and including both surgical and non-surgical treatments adds value. However, several issues need to be fixed before the paper can be accepted.

Major Comments

1. Only the PubMed search is clearly shown. The searches in Scopus, Web of Science, and ScienceDirect are not described well enough to replicate. PRISMA-ScR requires that at least one full search strategy be shown and that the others be clearly explained. Also, some sexual health terms are missing (dyspareunia, vaginal pain, libido, arousal, sexual desire). This may have caused missed studies. Please provide full search strings for all databases or include them in a supplement and expand the keywords.

2. Limiting the search to the last 10 years leaves out older but important PFMT and MUS studies that reported sexual outcomes. Because this is a scoping review, it should map all available evidence, not only recent studies. Please explain why these older studies were excluded.

3. The PRISMA flow diagram has inconsistent numbers. For example, “Records screened (n=3302)” in one place and “Records screened (n=300)” in another. Also, “Reports sought for retrieval (n=0)” contradicts the use of full-text screening. Some boxes are duplicated or mislabeled. The entire diagram needs to be redone with correct, consistent numbers.

4. You used MMAT and excluded studies scoring below 3/5. Scoping reviews usually do not exclude studies based on quality. The JBI approach recommends describing quality rather than using it to exclude studies. Either explain clearly why you excluded low-quality studies or re-include them and discuss their weaknesses in the results.

5. The paper reports very specific numbers (for exampe “32% improved, 13% worsened sexual function”) that come from only one study or a small group. These numbers should not be applied to all surgeries. PFMT and PMS are also described as “robust,” but evidence is mixed. Some RCTs show improvement; others show no change in FSFI. The statement that non-surgical treatments are “comparable” to surgery is not supported by head-to-head trials. The conclusions should be toned down. Link each number to its original study and avoid generalizing.

6. The introduction uses the WHO definition of sexual health, but the studies included only measure sexual function (FSFI, PISQ-12, GRISS). Broader sexual health topics were not assessed. Please clarify that this review focuses solely on sexual function outcomes.

7. The manuscript says partner outcomes show minimal change, but only 2–3 small studies measured partner sexual function. Sample sizes were small, and outcomes varied. Please state exactly how many partner-included studies there are, the total number of partners studied, and make it clear that the evidence is very limited.

8. The paper says, “no data available,” but the methods describe extracting data. Scoping reviews normally share extraction tables. Please make the data statement consistent and consider adding the extracted data as a supplementary file.

Minor Comments

There are several grammar mistakes and typos that need correction (e.g., “??” in the research question). Abbreviations need to be consistent. MUS and MUT are both used. Figures are low quality and should be remade with clearer labels. When discussing PFMT variability, include recent RCTs from 2020–2024 to support your points.

Reviewer #2: The authors aim to synthesize evidence on interventions to improve sexual health in women with Urinary Incontinence (UI), including the impact on partner satisfaction. While this is an interesting topic, the review has significant flaws, including signficnat heterogeneity and a lack of clear definitions, which makes conclusions hard to interpret.

Introduction: The introduction is lengthy and tries to cover too much at once. Furthermore, the aim is not specific; for example, it is unclear if the primary goal is to identify research gaps or to compare treatment approaches. The research question found in the methods section is much clearer than the aim stated in the introduction, and these two sections should be better aligned.

Methods/Results: While the methodology follows established guidelines, it fails to provide essential details on how sexual dysfunction was determined. It is not clear if the women in these studies had sexual dysfunction at baseline, which is a major limitation—it is difficult to study the improvement of a condition if the prevalence is unknown at the start. Additionally, while the authors mention using questionnaires like the PISQ and FSFI, they do not explain how these were used to draw conclusions or what was considered a "significant" change, especially in cases where no baseline dysfunction was recorded.

Additionally, regarding methodology, a major concern I have is grouping all Stress UI, Urge UI, and Pelvic Organ Prolapse together. These are biologically different conditions with unique treatment algorithms, and grouping them makes the results misleading. Because the surgical interventions mentioned are largely composed of prolapse repairs, they do not accurately represent UI treatment as a standalone category. This makes the comparison between surgical and non-surgical treatments invalid. On a positive note, the attempt to investigate partner satisfaction was interesting; the lack of significant reporting in this area serves as a helpful insight that more research is needed in this specific gap.

Discussion: The discussion claims that the findings have significant implications for clinical practice, but this feels like an overstatement that "oversells" the outcome. Overall, the review does not add much to the existing knowledge because its heterogeneous nature—including women with and without sexual dysfunction across different types of UI—makes it difficult to accurately assess the impact of treatment.

The paper would be much more insightful if, instead of qualitatively comparing treatments in a mixed population, it focused on why sexual dysfunction exists in each specific setting and/or describe and compare strategies within one category only. I would eliminate the comparison surgical and non-surgical treatment for UI since it’s not comparing the same group of patients, and I would encourage the authors to better define their study population (clear state if they all have UI, prolapse or sexual dysfunction at baseline).

6. PLOS authors have the option to publish the peer review history of their article (what does this mean?). If published, this will include your full peer review and any attached files.). If published, this will include your full peer review and any attached files.). If published, this will include your full peer review and any attached files.). If published, this will include your full peer review and any attached files.

...

Reviewer #1: No

Reviewer #2: **Yes:** Lina Posada CalderonLina Posada CalderonLina Posada CalderonLina Posada Calderon

---

## [Author Response · Author response to Decision Letter 1]

29 Jan 2026

Dear Dr. Lee,

Dear Reviewers,

We would like to thank the Academic Editor and both reviewers for their careful evaluation of our manuscript entitled “Strategies to improve sexual health in women with urinary incontinence: A scoping review” (PONE-D-25-66253). We appreciate the constructive and insightful comments, which have helped us substantially improve the methodological transparency, conceptual clarity, and clinical interpretation of our review.

Below, we respond point by point to each comment. Reviewer comments are reproduced verbatim in italics, followed by our responses and a description of the revisions made.

Responses to Reviewer #1

Comment 1

“Only the PubMed search is clearly shown… PRISMA-ScR requires that at least one full search strategy be shown and that the others be clearly explained… some sexual health terms are missing (dyspareunia, vaginal pain, libido, arousal, sexual desire).”

Response:

We thank the reviewer for highlighting this important methodological issue. We have now fully expanded the search strategy to ensure transparency and reproducibility in accordance with PRISMA-ScR guidelines.

• Full, reproducible search strings for PubMed, Scopus, Web of Science, and ScienceDirect are provided in Table1.

• We expanded the sexual health–related keywords to include dyspareunia, vaginal pain, libido, arousal, sexual desire, and related synonyms.

• The Methods section has been revised to clearly state how each database was searched and how terminology was adapted across platforms.

Sexual health–related terms were intentionally broadened to include dyspareunia, libido, sexual desire, arousal, and vaginal pain to minimize the risk of missing relevant studies

Comment 2

“Limiting the search to the last 10 years leaves out older but important PFMT and MUS studies… Please explain why these older studies were excluded.”

Response:

We appreciate this observation. We have now explicitly justified the time restriction in the Methods section. The rationale for limiting the search to the last 10 years was to:

1. Capture contemporary surgical techniques (e.g., modern mid-urethral sling modifications),

2. Reflect current PFMT protocols, and

3. Include recently validated sexual function instruments and scoring interpretations.

To address the reviewer’s concern, we now:

• Emphasize that this scoping review maps recent evidence trends, not the historical evolution of treatments.

The literature search was limited to studies published between January 2015 and December 2025. This timeframe was selected to capture evidence reflecting contemporary clinical practice in the management of urinary incontinence, including modern surgical techniques (e.g., current mid-urethral sling approaches), updated pelvic floor muscle training (PFMT) protocols, and the widespread use of validated sexual function instruments such as the FSFI and PISQ-12 with standardized scoring and interpretation. Given substantial changes in intervention techniques and outcome measurement over the past decade, restricting the timeframe was intended to enhance the clinical relevance and interpretability of the findings.

Comment 3

“The PRISMA flow diagram has inconsistent numbers… The entire diagram needs to be redone.”

Response:

We agree and thank the reviewer for identifying these inconsistencies. The PRISMA-ScR flow diagram has been completely redrawn.

Comment 4

“You used MMAT and excluded studies scoring below 3/5. Scoping reviews usually do not exclude studies based on quality.”

Response:

We thank the reviewer for this important methodological comment and fully acknowledge that scoping reviews typically do not exclude studies based on methodological quality. In this review, however, the application of a quality threshold served a specific preparatory purpose rather than a traditional evidence-synthesis aim.

The present scoping review was designed as a preliminary evidence-mapping step to inform the future development of a sexual health guideline for women with urinary incontinence. Given this applied objective, we prioritized studies with sufficient methodological rigor to ensure that the strategies identified would be based on reliable and interpretable evidence, suitable for translation into clinical or educational recommendations. Studies scoring below 3/5 on the MMAT frequently lacked clear outcome definitions, appropriate comparison groups, or valid sexual function measurement tools, which limited their utility for this purpose.

We have now explicitly clarified this rationale in the Methods section and emphasized that quality assessment was used to enhance the trustworthiness of mapped intervention strategies, rather than to rank effectiveness or exclude evidence arbitrarily. We also acknowledge this approach as a deviation from conventional scoping review methodology and have transparently discussed it as a methodological consideration in the manuscript.

Comment 5

“Very specific numbers… should not be applied to all surgeries… conclusions should be toned down.”

Response:

We fully agree. The Results and Discussion sections have been revised to:

• Avoid generalization across surgical modalities,

• Remove language suggesting equivalence or comparability between surgical and non-surgical interventions.

Terms such as “robust” and “comparable” have been replaced with more cautious, evidence-aligned wording.

Abstract, results:

The included 31 studies evaluated a broad range of interventions, including pelvic floor muscle training, behavioral therapies, pharmacological treatments, and surgical procedures. Reported effects on sexual function were inconsistent and varied across studies, depending on intervention type, study design, baseline sexual function status, and measurement tools. While some studies reported improvements in specific domains of sexual function following certain interventions, others reported no significant change or mixed outcomes. Evidence regarding partner-related sexual outcomes was limited and based on a small number of studies with modest sample sizes.

Abstract, conclusion:

This scoping review demonstrates that multiple interventions have been explored for their potential impact on sexual function in women with urinary incontinence; however, the available evidence is heterogeneous and largely descriptive. Due to variability in populations, interventions, and outcome assessment, no definitive conclusions regarding effectiveness or comparative benefit can be drawn. Future research using standardized sexual function measures and clearly defined populations is needed to better inform clinical practice and guideline development.

Discussion:

The review identified 31 studies. Improvements in sexual function outcomes were reported inconsistently across studies and varied by intervention type and outcome measure. Studies assessing pelvic floor muscle training (PFMT) demonstrated heterogeneous effects on sexual function. While some randomized controlled trials reported improvements in FSFI domains such as desire and satisfaction, other studies observed no statistically significant change following intervention. PFMT and PMS emerged as particularly robust, enhancing pelvic muscle tone and sexual satisfaction, while CBT addressed psychological barriers, improving sexual self-esteem [23, 25-28, 45]. Some studies about surgical interventions, including midurethral slings (MUS) and native tissue repairs, show enhanced sexual function and reduced coital incontinence, but some studies show sexual function deterioration[22, 23, 29-31, 35-40]. Non-surgical approaches were comparable to surgery for milder UI, offering less invasive options. However, interventions had minimal impact on partner sexual function, highlighting a patient-centric focus. These findings provide a comprehensive evidence base for tailored treatment strategies based on UI severity and patient preferences.

Comment 6

“The review uses the WHO definition of sexual health, but studies only measure sexual function.”

Response:

We appreciate the reviewer’s thoughtful comment regarding the conceptual distinction between sexual health and sexual function. The decision to frame this review within the broader construct of sexual health was intentional and grounded in contemporary sexual and reproductive health theory.

From a sexual health perspective, urinary incontinence affects women not only through measurable physiological sexual function domains (e.g., desire, arousal, orgasm), but also through psychological well-being, sexual self-esteem, intimacy, relationship dynamics, and fear or avoidance of sexual activity. Although many included studies operationalized outcomes using validated sexual function instruments such as the FSFI and PISQ-12, these tools capture multiple dimensions that extend beyond isolated physiological function and reflect broader aspects of sexual health as defined by the World Health Organization.

Moreover, as this scoping review was designed as a foundational step toward informing future sexual health–oriented guidance, adopting a sexual health framework allowed us to situate sexual function outcomes within a more comprehensive and clinically meaningful context. Nevertheless, to avoid conceptual ambiguity, we have now explicitly clarified throughout the manuscript that the empirical outcomes synthesized in this review relate primarily to sexual function measures, while the overarching conceptual framework remains grounded in sexual health.

Although sexual health is a multidimensional construct encompassing physical, psychological, and relational aspects of sexuality, the outcomes synthesized in this review primarily reflect sexual function as measured by validated questionnaires, which are commonly used proxies for sexual health–related outcomes in clinical research.

Comment 7

“Partner outcomes show minimal change, but only 2–3 small studies measured partner sexual function.”

Response:

We agree and have revised the manuscript to:

• Clearly state that evidence on partner outcomes is very limited and exploratory.

Finding:

Only three studies assessed partner-related sexual outcomes following interventions for urinary incontinence. Across these studies, with partner sample sizes that were either small and evaluated using heterogeneous outcome measures. Due to the limited number of studies, small sample sizes, and variability in outcome assessment, findings related to partner sexual outcomes should be interpreted with caution.

Comment 8

“The paper says ‘no data available,’ but scoping reviews normally share extraction tables.”

Response:

We appreciate this clarification. We have:

Revised the Data Availability Statement to ensure internal consistency.

Full data extraction was presented in table as Supplementary File S1, detailing study characteristics and outcomes.

Minor Comments

All typographical errors, inconsistent abbreviations (e.g., MUS vs. MUT), and figure quality issues have been corrected. Figures were remade following PLOS ONE technical guidelines, and recent RCTs (2020–2024) were added where relevant.

Responses to Reviewer #2

Introduction

“The introduction is lengthy… the aim is not specific.”

Response:

We have streamlined the Introduction and aligned the stated aim with the research question presented in the Methods section. The revised aim now clearly reflects the scoping purpose of mapping evidence and identifying gaps.

Introduction

Urinary incontinence (UI) is a common and distressing condition defined by the International Urogynecological Association as any involuntary leakage of urine resulting in social or health-related discomfort [1]. Although it affects both sexes, UI is more prevalent among women due to anatomical, hormonal, and obstetric factors. Global prevalence estimates range from 3% to 55% and increase substantially with age [2].

UI is classified into stress urinary incontinence (SUI), urgency urinary incontinence (UUI), mixed urinary incontinence (MUI), and overflow incontinence. SUI involves leakage during increased intra-abdominal pressure and is commonly associated with urethral hypermobility or intrinsic sphincter deficiency, whereas UUI is characterized by involuntary leakage accompanied by urgency and is often linked to detrusor overactivity [3,4]. MUI presents features of both, while overflow incontinence, the least common subtype in women, results from bladder overdistension due to obstruction or neurological impairment [5].

UI represents a significant public health concern because of its high prevalence, economic burden, and negative impact on physical and psychological well-being [6]. Increasing evidence suggests that UI is associated with female sexual dysfunction (FSD), likely due to the close anatomical and functional relationship between the urinary and reproductive systems. However, research examining this association remains limited and methodologically heterogeneous, resulting in inconsistent findings [7]. Coital incontinence, in particular, is a highly distressing symptom that substantially impairs sexual satisfaction and intimacy [8].

Beyond physical symptoms, UI is associated with adverse mental health outcomes, including depressive symptoms, reduced self-esteem, and social withdrawal [9,10]. These multidimensional consequences underscore the importance of incorporating sexual health considerations into patient-centered UI care [11]. Current management strategies include conservative, pharmacological, and surgical approaches [12]; however, sexual health outcomes are not consistently evaluated when assessing treatment effectiveness [13].

Despite growing recognition of the relationship between UI and sexual health, substantial gaps persist in sexual health service provision for affected women. Healthcare providers frequently report limited knowledge or discomfort in addressing sexual concerns related to UI, contributing to stigma and unmet needs [14]. These challenges are amplified in conservative cultural settings where discussions of sexual health are particularly sensitive [15,16]. Furthermore, there is a lack of structured, evidence-based strategies specifically aimed at improving sexual health outcomes in women with UI [15].

Existing literature has largely focused on UI symptom management, with comparatively little attention given to interventions targeting sexual well-being [17]. Previous systematic reviews, including that by Pinheiro Sobreira Bezerra et al. (2020), have primarily examined prevalence and associations between UI and sexual function rather than systematically mapping intervention strategies [7].

The aim of this scoping review was to systematically map and describe the available evidence on surgical and non-surgical interventions evaluated for improving sexual health–related outcomes, primarily sexual function, in women with urinary incontinence, and to identify key knowledge gaps, including the limited evidence on partner-related sexual outcomes.

Methods / Results

“: While the methodology follows established guidelines, it fails to provide essential details on how sexual dysfunction was determined. It is not clear if the women in these studies had sexual dysfunction at baseline, which is a major limitation—it is difficult to study the improvement of a condition if the prevalence is unknown at the start. Additionally, while the authors mention using questionnaires like the PISQ and FSFI, they do not explain how these were used to draw conclusions or what was considered a "significant" change, especially in cases where no baseline dysfunction was recorded.

Additionally, regarding methodology, a major concern I have is grouping all Stress UI, Urge UI, and Pelvic Organ Prolapse together. These are biologically different conditions with unique treatment algorithms, and grouping them makes the results misleading. Because the surgical interventions mentioned are largely composed of prolapse repairs, they do not accurately represent UI treatment as a standalone category. This makes the comparison between surgical and non-surgical treatments invalid. On a positive note, the attempt to investigate partner satisfaction was interesting; the lack of significant reporting in this area serves as a helpful insight that more research is needed in this specific gap."

R

---

## [Decision Letter · Decision Letter 1]

17 Mar 2026

Strategies to improve sexual health in women with urinary incontinence: A scoping review

PONE-D-25-66253R1

Dear Dr. Shahali,

We’re pleased to inform you that your manuscript has been judged scientifically suitable for publication and will be formally accepted for publication once it meets all outstanding technical requirements.

Kind regards,

Richard Kao Lee, M.D.

Academic Editor

PLOS One

Additional Editor Comments (optional):

Reviewers' comments:

Reviewer's Responses to Questions

**Comments to the Author**

1. If the authors have adequately addressed your comments raised in a previous round of review and you feel that this manuscript is now acceptable for publication, you may indicate that here to bypass the “Comments to the Author” section, enter your conflict of interest statement in the “Confidential to Editor” section, and submit your "Accept" recommendation.

Reviewer #1: All comments have been addressed

Reviewer #2: All comments have been addressed

2. Is the manuscript technically sound, and do the data support the conclusions?

Reviewer #1: Yes

Reviewer #2: Yes

3. Has the statistical analysis been performed appropriately and rigorously? 

Reviewer #1: Yes

Reviewer #2: Yes

4. Have the authors made all data underlying the findings in their manuscript fully available?

The PLOS Data policy requires authors to make all data underlying the findings described in their manuscript fully available without restriction, with rare exception (please refer to the Data Availability Statement in the manuscript PDF file). The data should be provided as part of the manuscript or its supporting information, or deposited to a public repository. For example, in addition to summary statistics, the data points behind means, medians and variance measures should be available. If there are restrictions on publicly sharing data—e.g. participant privacy or use of data from a third party—those must be specified.requires authors to make all data underlying the findings described in their manuscript fully available without restriction, with rare exception (please refer to the Data Availability Statement in the manuscript PDF file). The data should be provided as part of the manuscript or its supporting information, or deposited to a public repository. For example, in addition to summary statistics, the data points behind means, medians and variance measures should be available. If there are restrictions on publicly sharing data—e.g. participant privacy or use of data from a third party—those must be specified.requires authors to make all data underlying the findings described in their manuscript fully available without restriction, with rare exception (please refer to the Data Availability Statement in the manuscript PDF file). The data should be provided as part of the manuscript or its supporting information, or deposited to a public repository. For example, in addition to summary statistics, the data points behind means, medians and variance measures should be available. If there are restrictions on publicly sharing data—e.g. participant privacy or use of data from a third party—those must be specified.requires authors to make all data underlying the findings described in their manuscript fully available without restriction, with rare exception (please refer to the Data Availability Statement in the manuscript PDF file). The data should be provided as part of the manuscript or its supporting information, or deposited to a public repository. For example, in addition to summary statistics, the data points behind means, medians and variance measures should be available. If there are restrictions on publicly sharing data—e.g. participant privacy or use of data from a third party—those must be specified.

Reviewer #1: Yes

Reviewer #2: Yes

5. Is the manuscript presented in an intelligible fashion and written in standard English?

Reviewer #1: Yes

Reviewer #2: Yes

6. Review Comments to the Author

Reviewer #1: The authors have appropriately addressed all of my comments and revisions. I do not have any additional comments at this time.

Reviewer #2: The authors have spend a significant time addressing and reviewing all the comments. The studies strengths and limitations and more clear. This paper does add to the literature by providing a comprehensive literature review of sexual outcomes in patients managed for urinary incontinence.

7. PLOS authors have the option to publish the peer review history of their article (what does this mean?). If published, this will include your full peer review and any attached files.). If published, this will include your full peer review and any attached files.). If published, this will include your full peer review and any attached files.). If published, this will include your full peer review and any attached files.

...

Reviewer #1: No

Reviewer #2: No

---

## [Editor Report · Acceptance letter]

PONE-D-25-66253R1

PLOS One

Dear Dr. Shahali,

I'm pleased to inform you that your manuscript has been deemed suitable for publication in PLOS One. Congratulations! Your manuscript is now being handed over to our production team.

Kind regards,

on behalf of

Dr. Richard Kao Lee

Academic Editor

PLOS One